# Resident Esophageal Microbiota Dysbiosis Correlates with Cancer Risk in Barrett’s Esophagus Patients and Is Linked to Low Adherence to WCRF/AICR Lifestyle Recommendations

**DOI:** 10.3390/nu15132885

**Published:** 2023-06-26

**Authors:** Alice Zaramella, Diletta Arcidiacono, Daniele Nucci, Federico Fabris, Clara Benna, Salvatore Pucciarelli, Matteo Fassan, Alberto Fantin, Vallì De Re, Renato Cannizzaro, Stefano Realdon

**Affiliations:** 1Department of Surgery, Oncology and Gastroenterology (DiSCOG), University of Padua, Via Giustiniani 2, 35128 Padua, Italy; clara.benna@unipd.it (C.B.); puc@unipd.it (S.P.); 2Gastroenterology Unit, Veneto Institute of Oncology IOV-IRCCS, Via Gattamelata 64, 35128 Padua, Italy; diletta.arcidiacono@iov.veneto.it (D.A.); alberto.fantin@iov.veneto.it (A.F.); 3Dietetics and Clinical Nutrition Unit, Veneto Institute of Oncology IOV-IRCCS, Via Gattamelata 64, 35128 Padua, Italy; daniele.nucci@iov.veneto.it; 4Department of Biomedical Sciences, University of Padua, Viale Colombo 3, 35121 Padua, Italy; federico.fabris.5@phd.unipd.it; 5Department of Medicine (DIMED), University of Padua, Via Gabelli 61, 35121 Padua, Italy; matteo.fassan@unipd.it; 6Veneto Institute of Oncology IOV-IRCCS, Via Gattamelata 64, 35128 Padua, Italy; 7Immunopathology and Cancer Biomarkers, Centro di Riferimento Oncologico di Aviano (CRO), National Cancer Institute, IRCCS, 33081 Aviano, Italy; vdere@cro.it; 8Oncological Gastroenterology, Centro di Riferimento Oncologico di Aviano (CRO), National Cancer Institute, IRCCS, 33081 Aviano, Italy; rcannizzaro@cro.it (R.C.); stefano.realdon@cro.it (S.R.); 9Department of Medical, Surgical and Health Sciences, University of Trieste, 34127 Trieste, Italy

**Keywords:** microbiota, esophageal adenocarcinoma, diet

## Abstract

Esophageal adenocarcinoma (EAC) is the consequence of longstanding gastroesophageal reflux, which leads to inflammation and could cause Barrett’s esophagus (BE), the main risk factor for EAC development. The 5 year survival rate of EAC is poor since the diagnosis occurs at the late stage of the disease. To improve patient management, a better comprehension of the mechanism undergoing the evolution through to adenocarcinoma is needed. Within this scenario, the resident microbiome investigation was studied. This study aimed to explore the esophageal microbial profile in patients affected by non-dysplastic BE, low- and high-grade dysplastic BE, and EAC to identify parameters characterizing cancer progression and to develop a score suitable for clinical practice to stratify cancer risk. The microbiota was investigated through the 16S rRNA gene sequencing of esophageal biopsies. The microbial composition was evaluated at each different taxonomic level along the disease progression. To further investigate bacteria potentially associated with cancer development, non-dysplastic and dysplastic/cancer patients were compared. The presence of the six significant microbial features with multivariate analysis was used to develop a multiparametric score (Resident Esophageal Microbial Dysbiosis Test) to predict the risk of progression toward EAC. Finally, the diagnostic ability of the test and its discrimination threshold for its ability to identify dysplastic/cancer patients were demonstrated. Since EAC has been related to obesity, the relationship between these microbial parameters and patients’ diet/lifestyle habits was also investigated. Developing microbiome-based risk prediction models for esophageal adenocarcinoma onset could open new research avenues, demonstrating that the resident microbiome may be a valid cancer risk biomarker.

## 1. Introduction

Esophageal cancer is now the seventh malignancy in terms of incidence in the worldwide ranking, and the sixth in mortality [1]. Esophageal adenocarcinoma (EAC) and squamous cell carcinoma (ESCC) are the two most frequent histological types [2]. Although the incidence of ESCC has been declining in the last few years, the same is not true for EAC, especially in Western countries [2]. The age of EAC onset is between 50 and 60 years, and it is more frequent in males. EAC is usually asymptomatic until advanced disease stages, when symptoms appear, among which dysphagia and weight loss are more common. EAC often develops after a longstanding gastroesophageal reflux disease (GERD), and its principal precursor is the presence of Barrett’s esophagus (BE), which is a specialized intestinal metaplasia of the epithelium of the lower esophagus. BE is considered the main risk factor since most cases of EAC arise in the background of BE [3]. GERD, BE, and EAC are closely linked, defining a sequence that begins with GERD, progresses with BE development, and, through the intermediate step of dysplasia (first low-grade and then high-grade), ends with EAC development [4]. A better understanding of the key features of the mechanism in progress during the evolution through to EAC is needed and could ameliorate patient management as well as improve cancer risk stratification.

Although obesity and higher values of waist circumference are well-established factors that modulate the progression through said sequence [5], the role of diet and lifestyle habits is not well defined. However, evidence highlights the role of dietary habits in the development of cancer, especially regarding the consumption of animal-derived proteins and high-glycemic-index foods [6,7,8]. Moreover, in 2007 the World Cancer Research Fund (WCRF) and the American Institute of Cancer Research (AICR) published a series of recommendations on diet, physical activity, and weight management for cancer prevention, based on the most comprehensive collection of available evidence [9]. The concordance with the guidelines was proven to be associated with reduced mortality from different cancers, including EAC [10,11,12]. 

In recent years, new data have suggested the possible role of resident gastroesophageal microbiota in the progression from pre-neoplastic lesions to adenocarcinoma [13,14,15,16]. The introduction of high-throughput sequencing technologies (NGS) has confirmed the existence of a typical microbial community inhabiting the human esophagus and stomach, which differs from those shown in the oral cavity and the bowel [17]. Ever since the early studies investigating esophageal microbiome composition, it has been clear that *Streptococcus*, *Prevotella*, and *Veillonella* were the most represented in order of abundance [15,18,19]. The composition could be influenced by age, drugs (especially proton pump inhibitors), and dietary habits [20,21,22]. Healthy microbial composition, principally by Gram-positive bacteria (Firmicutes phylum), was defined as Type I in the work of Yang and colleagues [15], suggesting the shift into a Type II during esophageal disease. Type II is dominated by Gram-negative taxa in patients with both GERD and BE. 

This work aimed to investigate esophageal microbiota in patients affected by non-dysplastic BE, BE with dysplastic lesions (low and high-grade dysplasia), and BE with cancer (EAC), and evaluated the association between its main features and dietary/lifestyle habits during EAC development. The main goal was to identify the most representative microbial parameters and to develop a multiparametric score able to predict the risk of progression toward EAC in BE patients to improve patient management. Possible associations between the esophageal microbial community and adherence to WCRF/AICR recommendations were also investigated.

## 2. Materials and Methods

### 2.1. Patients and Samples Collection

Consecutive patients undergoing upper endoscopy from 1 October 2014 to 1 October 2017 at the Digestive Endoscopy Unity of the IOV-IRCCS with an already-known diagnosis of BE and who were, therefore, included in a follow-up endoscopic protocol, were considered. For each patient, during endoscopy, biopsies were taken following the Seattle protocol and, if areas suggestive of dysplasia or cancer were visualized (i.e., pit pattern distortion under NBI vision and/or white light macroscopically visible lesions), target biopsies were taken as well. All specimens were fixed in formalin and examined by pathologists. Information on the esophageal microbial profile was obtained from additional target biopsies collected from the distal esophagus and immediately frozen in liquid nitrogen. Biopsy specimens were stored at −80 °C within the Surgical Biobank of Padua’s University. The study was conducted in accordance with the Declaration of Helsinki, in compliance with good clinical practice and local regulations. Informed consent was obtained from all the patients, and all the information was recorded anonymously according to the regulations of our institution. This study was approved by the Ethics Committee of our Institution (CE-IOV 2014/67).

The exclusion criteria were the following:(i)Presence of other types of cancer;(ii)Taking antibiotics/probiotics within 3 months;(iii)Previous esophageal surgery.

According to histological findings, patients were divided into four groups: patients with non-dysplastic BE, patients with BE and low-grade dysplasia (LGD), patients with BE and high-grade dysplasia (HGD), and patients with BE and EAC. 

### 2.2. Lifestyle and Diet Questionnaire

At the time of the endoscopic examination, a questionnaire developed reflecting the WCRF/AICR guidelines on cancer prevention [11,23] was administered to each patient. The applicable recommendations to our population were used for the development of the questionnaire following our previous findings [12]: physical activity, consumption of foods that promote weight gain, consumption of drinks that promote weight gain, consumption of plant foods, and consumption of alcoholic drinks. Some recommendations had several sub-recommendations. To each component, a score of 1 was assigned when the recommendation was met and a score of 0 was assigned when it was not. To appraise a higher proportion of the variability in the population, an intermediate category was created and was given a score of 0.5. A final single score for each patient was obtained from the sum of each component. Higher scores indicated greater concordance with WCRF/AICR recommendations, as described in Table 1.

### 2.3. DNA Extraction and Sequencing

The esophageal microbiota profile was assessed on the additional biopsies obtained during endoscopy. Biopsies were mechanically destroyed through high-speed shaking with beads using Tissue Lyzer II (Qiagen, Hilden, Germany). After, bacterial DNA was extracted with QIAamp DNA Microbiome Kit (Qiagen, Hilden, Germany) following the manufacturer’s instructions. DNA concentration was determined using the Qubit dsDNA HS Assay kit and Qubit 2.0 Fluorometer (Thermo Fisher Scientific, Waltham, MA, USA). Purified bacterial DNA was amplified by targeting the V3-V4 region of the bacterial 16S rRNA gene. Libraries were prepared using the QIAseq 16S region panel for the V3-V4 region (Qiagen, Hilden, Germany), following the manufacturer’s instructions. Sequencing libraries were labeled with different multiplex indexing barcodes for each sample. The presence of the target sequences was evaluated with the Agilent TapeStation (Santa Clara, CA, USA) using the D1000 ScreenTape and Reagents. Libraries’ quantifications were assessed using the QIAseq Library Quant (Qiagen, Hilden, Germany). Amplicons were sequenced using the Illumina Miseq platform (Miseq Reagent Kit v3 600 cycles, Illumina, CA, USA). Finally, sequences were trimmed, filtered, merged, and clustered into operational taxonomic units (OTUs) using CLC Genomics Workbench and CLC Microbial Genomics Module v.21 (Qiagen, Hilden, Germany). 

### 2.4. Bioinformatic Analysis

The 16SrRNA raw sequences dataset, generated from the Miseq run, was merged, demultiplexed, trimmed down to 250 nucleotides, and filtered. High-quality filtered reads were clustered into operational taxonomic units (OTUs) using CLC Genomics Workbench and CLC Microbial Genomics Module v.21 (Qiagen, Germany). The taxonomic assignment of sequences was carried out based on the SILVA (version 132) database with 97% of similarity. Clusters of OTUs composed of only 1 read were discarded, and OTUs belonging to Eukarya, Archea, Chloroplast, Cyanobacteria, and Mithocondria were removed from downstream analysis. Samples’ biodiversity (alpha-diversity) was estimated according to different microbial metrics such as Shannon, Chao-1 indices, and Faith’s Phylogenetic distance (FD). Moreover, inter-sample diversity (beta-diversity) was calculated using both Weighted and Unweighted Unifrac matrices and visualized as a principal coordinate analysis plot (PCoA). Alpha and beta analyses were performed using CLC Microbial Genomics Module v.21 (Qiagen, Germany).

### 2.5. Statistical Analysis

Alpha diversity indices were analyzed using one-way analysis of variance (ANOVA) with the false discovery rate (FDR) correction. Since beta diversity indices are matrices, the multi-group comparison of data was carried out through the Permutational Multivariate Analysis of Variance (PERMANOVA). The Kruskal–Wallis test (KW) or the one-way ANOVA analysis of variance was used for multiple comparisons. Cuzick’s test for trend was used to measure the trend of microbial relative abundance along the disease progression. The Mann–Whitney U test (MW) was used to assess differences between pre-cancerous patients and patients who progressed through cancer. For correlation analysis, the Spearman’s rank correlation test was performed. To estimate the association between microbiota features and the disease state, variables significantly different between patients with pre-neoplastic lesions and patients who progressed through dysplasia or cancer were subjected to univariable logistic analysis. Those with a *p* value < 0.05 were incorporated into the multivariable logistic analysis to determine the risk factors for cancer progression in pre-cancerous patients. The receiver operating characteristic (ROC) curve was used to analyze the critical values of the microbiota features, and therefore sensitivity, specificity, positive predictive values, and negative predictive values were also obtained. A *p* value lower than 0.05 was assumed to indicate a significant difference. Data analyses were performed using STATA 12.0 (StataCorp. 2011. Stata Statistical Software: Release 12. College Station, TX: StataCorp LP, USA) and GraphPad Prism9 (GraphPad Software Inc., La Jolla, CA, USA).

## 3. Results

### 3.1. Characteristics of the Study Population

According to histological findings, a total of 80 consecutive patients were included in our cohort. Patients were divided into four groups: 57 patients with non-dysplastic BE (BE group), 8 patients with low-grade dysplastic BE (LGD group), 8 patients with high-grade dysplasia (HGD group), and 7 patients with esophageal adenocarcinoma (EAC group). The characteristics of the population are reported in Table 2.

### 3.2. Quality of Sequencing and Diversity Measurements throughout the Disease

The V3-V4 sequencing of the 16S rRNA gene on the distal esophagus samples resulted in 5,315,292 high-quality reads, with an average of 66,441 reads per sample available for the microbiota analysis. After the filtration and removal of chimeric reads, 2,258,379 reads in OTUs were obtained. A total of 3517 OTUs were found. The number of OTUs was not significantly different among groups (ANOVA, *p* = 0.643), as shown in Figure 1a. Alpha diversity was analyzed as the measure of within-sample diversity. The Chao1, Shannon, and Faith’s FD indices were used to describe the alpha diversity of esophageal microbiota for each group studied in esophageal biopsies. Neither the richness (Chao1 index) nor the diversity (Shannon index) of the microbiota showed significant differences among groups (Kruskal–Wallis, *p* = 0.321; *p* = 0.536, respectively), which was also shown in Faith’s FD. The data are reported in Figure 1b–d. To assess the diversity between studied groups, beta diversity was calculated using both the Bray–Curtis and Unweighted UniFrac phylogenetic distance matrices, and shown in the PCoA plots (Figure 2). However, the PERMANOVA analysis of beta diversity did not show significant differences among groups for both indices calculated (*p* = 0.074 and *p* = 0.323, respectively).

### 3.3. Differences in Microbiota Composition throughout the Disease

The composition of the bacterial community inhabiting the esophagus of patients was evaluated at each different taxonomic level in the distal esophageal biopsies along the BE–LGD–HGD–EAC sequence. The microbiota composition was dominated by the Firmicutes phylum in each stage of the disease, followed by Proteobacteria, Bacteroidetes, Actinobacteria, and Fusobacteria in descending order of abundance. Taken together, their percentage reached 90% of the total microbial composition of the esophageal mucosa. According to the results obtained in a previous study [15], the healthy esophageal microbial composition should be dominated (>60%) by the Firmicutes phylum, mostly by the *Streptococcus* genus. In our cohort, this percentage was between 41.0%, and 35.0% without significant differences in different groups, as shown in Figure 3a. The Bacteroidetes phylum (composed of only Gram-negative bacteria) was higher in EAC patients, but this difference did not reach statistical significance (KW, *p* = 0.333). Conversely, the Kruskal–Wallis test showed a significant difference in Patescibacteria abundance among groups (KW, *p* = 0.004). The post hoc analysis showed a higher abundance of this phylum in LGD patients (*p* = 0.002) (Figure 3b), compared to BE. The complete list of principal phyla observed in the distal esophagus of our patients is reported in detail in Appendix A. Within the most abundant phyla, forty-nine different bacterial classes were distinguished. The most abundant classes were those belonging to Firmicutes (Bacilli, Negativicutes, and Clostridia classes), Proteobacteria (Gammaproteobacteria and Alphaproteobacteria classes), Bacteroidetes (Bacteroidia class), Fusobacteria (Fusobacteriia class), and Actinobacteria (Actinobacteria class) phyla (Appendix A). The Kruskal–Wallis test showed no differences in the relative abundance of the main bacterial classes among the four groups of patients. Conversely, considering the less abundant classes, the analysis showed significant differences in Gracilibacteria (belonging to the Patescibacteria phylum, KW test result *p* < 0.001). Bacteria belonging to Gracilibacteria class were more abundant in LGD patients. Post hoc tests showed significant differences between the LGD and the BE group (*p* < 0.001) (Figure 3c). 

Within the most abundant classes, sixty different bacterial orders were distinguished. The most abundant orders belonged to Bacilli (Lactobacillales and Bacillales orders), Gammaproteobacteria (mostly Pasteurellales, Betaproteobacteriales, and Pseudomonadales orders), Bacteroidia (Bacteroidales and Flavobacteriales orders), Alphaproteobacteria (mostly Sphingomonadales and Rhizobiales orders), Negativicutes (Selenomonadales order), Fusobacteriia (Fusobacteriales order), Actinobacteria (mostly Micrococcales and Actinomycetales orders), and Clostridia (Clostridiales order) classes. The most abundant orders (relative abundance > 0.5%) are shown in Figure 4a; the complete list, including less abundant bacteria, is reported in Appendix A. 

The Cuzick test for the trend showed that the relative abundance of bacteria belonging to the Bacteroidales order (composed of Gram-negative bacteria) was significantly increased along the disease progression (Cuzick test result, z = +2.13, *p* = 0.033, Figure 4b), with a median value from 15.0% in BE to 25.0% in EAC patients. 

Within the most abundant orders, sixty-six different bacterial families were distinguished. The most abundant families belonged to Lactobacillales (mostly *Streptococcaceae* family), Bacteroidales (mostly *Prevotellaceae* and *Porphyromonadaceae* families), Pasteurellales (*Pasteurellaceae*), Selenomonadales (*Veillonellaceae*), Micrococcales (*Micrococcaceae*), Actinomycetales (*Actinomycetales*), Fusobacteriales (*Fusobacteriaceae* and *Leptotrichiaceae*), and Clostridiales (*Lachnospiraceae* family) orders. The detailed list of bacterial families is reported in Appendix A. 

The most abundant genera (relative abundance >0.5%) are shown in Figure 4c; the complete list, including less abundant bacteria, is reported in Appendix A. Within the *Prevotellaceae* family, the *Alloprevotella* genus increased throughout the disease progression (Cuzick’s trend test z = +2.03; *p* = 0.043; Figure 4d). Moreover, a significant increase was observed in the abundance of some Gammaproteobacteria, among which were *Eikenella* and *Aggregatibacter* genera (Cuzick’s trend test z = +3.15, *p* = 0.002; z = +2.66, *p* = 0.008, respectively), as reported in Figure 4e,f.

### 3.4. Main Distinguishing Features between Non-Dysplastic BE Patients and Patients with BE-Related Dysplasia or Cancer

To identify bacteria potentially associated with cancer development, non-dysplastic BE patients (n = 57) and dysplastic/EAC patients (LGD, HGD, EAC together, n = 23) were compared. The analyses allowed us to identify which bacteria characterized the shift in resident esophageal microbiota during carcinogenesis. As shown by the PERMANOVA results performed comparing non-dysplastic BE patients and the dysplastic/EAC group, the differences in microbial composition between the two groups were significant (PERMANOVA test results: f-statistic = 2.01 *p* = 0.014; f-statistic = 2.98, *p* = 0.011 for the Bray–Curtis and Unweighted UniFrac, respectively). In the PCoA graph, two distinct populations were distinguished, indicating the clear differentiation of two clusters. Data are shown in Figure 5. 

The relative abundances of bacteria at each taxa level are reported in Appendix A. A higher abundance of bacteria belonging to the Firmicutes’ phylum was observed in BE compared to the progression group (41.1% and 36.7%, respectively). However, this difference did not reach statistical significance (MW test result *p* = 0.255). A significantly higher abundance of bacteria belonging to the Gracilibacteria class (Patescibacteria phylum) (MW, *p* = 0.002) was observed in patients who progressed (Figure 6a,b). 

At the family level, the largest percentage of bacteria belonged to *Streptococcaceae* (26.0% and 22%, the median values in BE and the progression group, respectively), *Prevotellaceae* (14.0% and 17.0%, respectively), *Pasteurellaceae* (7.0% and 10.0%, respectively), *Veillonellaceae* (6.0% for both groups), *Micrococcaceae* (2.0% for both groups), *Fusobacteriaceae* (1.0% and 2.0%, respectively), *Neisseriaceae* (1.0% and 2.0%, respectively), and *Sphingomonadaceae* (1.0% and 3.0%, respectively) families. As shown in Figure 6c, a higher abundance of bacteria belonging to *Lachnospiraceae* (Firmicutes phylum) was observed in BE patients compared to the progression group (MW, *p* = 0.046, *p* = 0.038, respectively). Moreover, a significant reduction of *Burkholderiaceae* and *Rhizobiaceae* (Proteobacteria phylum) (MW, *p* = 0.002, *p* = 0.018, respectively) was also observed (Figure 6d,e). 

At the genus level, the largest percentages of bacteria were of *Streptococcus* (26.0% and 22.0%, the median values in BE and progressed patients, respectively), *Prevotella* (7.0% and 10.0%, respectively), *Haemophilus* (6.0% for both groups), *Veillonella* (5.0% for both groups), *Alloprevotella* (3.0% and 4.0%), *Gemella* (2.0% and 3.0%), *Porphyromonas* (2.0% for both groups), *Actinomyces* (1.0% for both groups), *Rothia* (2. 0% and 1.0%), Fusobacteria (1.0% and 2.0%), *Neisseria* (0.9% and 2.0%), and *Sphingomonas* (0.9% and 3.0%). The percentages of *Bergeyella* and *Allopreovetella* genera (Bacteroidetes phylum) were significantly higher in the progression group (Figure 6f,g). Within Proteobacteria, the *Aggregatibacter* result was significantly higher in the progression group (MW, *p* < 0.001, Figure 6h). On the other hand, a decreased amount of *Acinetobacter* and *Massilia* (all belonging to Proteobacteria) was observed (Figure 6i,j).

### 3.5. Development of an Esophageal Microbiota Dysbiosis Test

After the quantification of the relative abundance of phyla, orders, classes, families, and genera and the assessment of the differences between non-dysplastic BE and patients who progress through EAC, further analyses were addressed to identify the bacteria potentially associated with cancer development that could be used as hallmarks of progression. To predict the dependence and relationship between microbial characteristics and disease progression, the logistic regression analysis (univariate) was performed. To determine which cut-offs would be used to discriminate BE patients from the progression group, we decided to subdivide the whole data distribution based on the quartiles. Therefore, the whole data distribution was divided into four quartiles: the first, quartile 1 (Q1), was the 25% percentile, the second, quartile 2 (Q2), was the 50% percentile, as the median of the dataset, the third, quartile 3 (Q3), was the 75% percentile, and quartile 4 (Q4) included the highest value. Accordingly, the odds ratio (OR) in the logistic regression was calculated to identify the percentage of patients with a relative abundance that was lower (or higher) compared to the cut-offs. However, just considering the relative abundance of bacteria discussed so far, no significant associations were found between the two groups (BE and progression group). These results convinced us to set up further analyses, considering the contribution of each class, order, family, or genus within a higher taxonomic level of belonging, as already suggested by other works [24,25,26]. For example, the *Streptococcus* (genus) relative percentage within its phylum, and the total Firmicutes relative percentage, was calculated as the ratio *Streptococcus*/Firmicutes. The alternative was calculating the ratio between orders (or families or classes) within the phylum of belonging, for example, o_Sphingomonadales/o_Rhizobiales, both included in the Alphaproteobacteria class and Proteobacteria phylum. Thus, the huge amount of data obtained was filtered according to the significance level of the statistical analyses and only significant results were considered. Figure 7 schematically reports these ratios, considered as “risk conditions”, which significantly increased or decreased during cancer development, as shown by the Mann–Whitney test results. The 10 significant ratios considered “risk conditions” were used to perform the logistic regression analysis. Data and significance levels were reassumed in Table 3. The ten risk conditions were successively considered for the multivariate analysis. Six out of ten risk conditions remained significant and significantly differentiated non-dysplastic BE patients from the progression group, and they are reported in Table 3. The number of risk conditions for each patient is reported in Figure 8a–c. A total of 98% of BE patients (56 out of 57) satisfied fewer than three risk conditions, and no BE met more than four risk conditions. Six out of eight LGD patients (75%) showed three (four patients) or four (two patients) risk conditions. A total of 62.5% of HGD patients showed more than four risk conditions. No patients with dysplastic or neoplastic lesions satisfied fewer than two risk conditions. Accordingly, the aforementioned classification was called the “Resident Esophageal Microbiota Dysbiosis Test” (REM-DT), and its value for each patient was calculated as the sum of the number of risk conditions. The REM-DT was significantly higher in the progression group compared to BE patients, as shown by the Mann–Whitney U test result reported in Figure 8d (the mean REM-DT in BE was 1.7 ± 0.8, and in the progression group it was 3.7 ± 1.1, *p* < 0.0001). To explore the criterion validity of the REM-DT, the ROC analysis was performed to determine the optimal cut-off to identify patients who progressed through cancer from non-dysplastic BE patients (Figure 8c). The area under the curve (AUC) was 0.92 (± 0.03), and a cut-off of ≥ 3 risk conditions yielded the best combination of sensitivity (91.30%), specificity (82.46%), a negative predicted value of 96.9%, and an accuracy of 85.0%. Applying this cut-off, 85.0% of patients were correctly classified. Considering BE patients that showed fewer than two risk conditions, to date 96.2% of patients are disease-free and 3.8% progressed to low-grade dysplasia in a mean follow-up period of 48 months. Furthermore, within the same follow-up period, 19.4% of BE patients presenting more than two risk conditions progressed to low-grade or high-grade dysplasia, and 80.6% were disease-free.

### 3.6. Adherence to WCRF/AICR Recommendations and Correlation with Microbiota Composition

The questionnaire was administered to each patient before the endoscopy. The complete list of results is reported in Figure 9c. Next, every single item was added up to calculate a total score for adherence to WCRF/AICR recommendations. As shown in Figure 9a, no differences were found between the median total score of patients in the studied groups (Kruskal–Wallis *p* = 0.263). Considering each item separately, the adherence to the consumption of vegetables and/or fruits (fruit item) was different among the groups (Kruskal–Wallis *p* = 0.048). In particular, the score for adherence was higher in BE compared to EAC patients (Figure 9b).

The Spearman’s Rank Correlation Coefficient was calculated to evaluate the strength of the association between adherence to WCRF/AICR suggestions and microbial parameters. A negative and significant correlation was observed between physical activity and the abundance of the most abundant Gram-negative phyla, such as Bacteroidetes, Proteobacteria, and Epsilonbacterota (r_s_ = −0.43, *p* < 0.001; r_s_ = +0.28, *p* = 0.011; and r_s_ = −0.24, *p* = 0.035, respectively). Moreover, a negative correlation was observed between other Gram-negative bacteria belonging to the *Veillonellaceae* family, in the Clostridia class (r_s_ = +0.24, *p* = 0.031). 

A significantly negative correlation was observed between the fruit item and Gram-negative bacteria such as Selenomonadales and Negativicutes orders, and in particular the *Veillonella* genus (Figure 10a–c). Conversely, a weak but positive correlation was observed with the *Porphyromonas* genus abundance, as shown in Figure 10d (r_s_ = +0.22, *p* = 0.047). Similarly, other Gram-negative bacteria were negatively correlated with the consumption of unprocessed plant food, such as Fusobacteria and Tenericutes (especially Mollicutes class) phyla (Figure 10e–g), the Betaproteobacteria class and the *Neisseria* genus (Figure 10h–i). Moreover, as shown in Figure 10j, the *Tannerella* genus (Bacteroidia phylum) was weakly correlated with the consumption of unprocessed plant food. A negative and significant correlation with fruit consumption was also observed in some Gram-positive bacteria, including in the Bacillales order and *Peptostreptococcaceae* (Clostridiales order) (r_s_ = −0.28, *p* = 0.011; r_s_ = −0.31, *p* = 0.006, respectively). Data are reported in Figure 10k, l. Remaining within Gram-positive bacteria, the fruit item was also positively correlated with the *Bifidobacteriaceae* family (Figure 10m). The adherence to the item red meat, which translates into a minor consumption of red meat, as suggested by recommendations, was again negatively correlated with Gram-negative bacteria. Figure 10n–q show the weak but significant correlation between this item and Fusobateriales and Clostridiales orders (r_s_ = −0.27, *p* = 0.013; r_s_ = −0.28, *p* = 0.017, respectively), and *Haemophilus* and *Leptotrichia* genera (r_s_ = −0.23, *p* = 0.044; r_s_ = −0.29, *p* = 0.008, respectively). Conversely, a significant and positive correlation was observed with the *Rhizobiaceae* family (Figure 10r). Other positive, weak but significant correlations were observed between the adherence to the recommendation regarding the limit of alcohol consumption and the abundance of Gram-positive bacterial genera, such as *Actinomyces*, *Granulicatella,* and *Rothia* (Figure 10s–u). Moreover, a weak but significant negative correlation was observed between alcohol and the *Bergeyella* genus (Flavobacteriia class, Figure 10v).

Ultimately, the strength of the association between each significant microbial parameter used to define the REM-DT and the adherence to WCRF/AICR suggestions was investigated. As shown in the correlation matrix (Figure 11), the limitation of sedentary habits and processed cereal consumption was weakly but significantly and positively correlated with the Bacilli/Clostridia ratio (r_s_ = −0.25, *p* = 0.026; r_s_ = −0.24, *p* = 0.031). Conversely, the *Bifidobacterium*/Actinobacteria ratio was weakly but significantly negatively correlated with both the eating of vegetables and/or fruits every day and the physical activity item (r_s_ = −0.27, *p* = 0.043; r_s_ = −0.23, *p* = 0.041). Similarly, the habit of physical activity was also negatively correlated with the *Aggregatibacter*/*Pasteurellaceae* ratio (Figure 11). 

## 4. Discussion

Esophageal adenocarcinomas develop through a cascade of pre-cancerous lesions, starting with Barrett’s esophagus. From that point, a series of dysplastic lesions occur that may progress through to cancer. To better identify early lesions, it is important to stratify patients with a higher risk of progression to personalize follow-up timing and strategy. A deeper knowledge of the transition from the normal epithelium to a dysplastic one could represent an additional method to stratify cancer risk, improving diagnostic procedures within a more cost-effective screening and follow-up protocol. Within this scenario, microbiota investigation has taken hold, due to its crucial role in health, as well as in other gastrointestinal diseases and cancers [25,27]. 

It is now accepted that the esophageal microbiome is altered during esophageal carcinogenesis. The well-noted risk factors associated with EAC also appear to be related to modifications to the normal microbiota that inhabits the esophagus [28]. Diets rich in high-fat contents in animal studies have been linked with esophageal dysplasia and alterations of the microbiota [29]. The low intake of fibers was also associated with an increase in Gram-negative species, which can be found in esophageal cancer precursor lesions [22,28]. Overall, the microbiota of the normal esophagus is mainly composed of Firmicutes, Proteobacteria, Bacteriodetes, Actinobacteria, and Fusobacteria [30]. In contrast, the esophageal cancer microbiota is characterized by a shift from the so-called Type I to the Type II microbiome, which corresponds to a shift from a condition with a high abundance of Gram-positive bacteria to an increase in Gram-negative bacteria, accompanied by reduced microbial diversity [15,16,31].

Our work focuses on analyzing the resident esophageal microbiota composition in patients affected by both pre-cancerous and cancerous lesions to identify possible pathogenetic-specific patterns during carcinogenesis.

Firstly, we compared the number of OTUs obtained with sequencing and investigated the alpha and beta indices. The number of OTUs and the alpha and beta diversity indices were not different among groups, as has already been shown by other works [14,32]. These data suggest that the within-sample diversity was not different among groups and therefore, like in a previous study [15], the shift in microbial composition from a healthy Type I microbiome had already begun at the BE condition, as shown by the low percentage (below the optimal 60%) of *Streptococcus* in BE patients. The identification of a healthy esophageal microbiome (Type I) composed of Gram-positive bacteria, in particular with a percentage of *Streptococcus* (Firmicutes phylum) higher than 60%, was proposed by Yang’s work in 2009 [15]. Type I could gradually shift through Type II (characterized by the presence of Gram-negative bacteria) in the context of esophageal diseases. What is most surprising is that the beta-diversity assessment did not show a significant different clusterization of the groups, suggesting a similar phylogenetic composition between BE, LGD, HGD, and EAC. To a certain extent, these data were confirmed by exploring the microbial composition at each different taxonomic level among each group. Although the Firmicutes amount was similar between groups, a significant increase in the Gram-negative order Bacteroidales (Bacteroidetes phylum) was observed along the disease progression. Since it has been noted that the Firmicutes phylum should be predominant in normal esophageal mucosa, an increased amount of Bacteroides (normally inhabiting the intestine) could represent the first step in esophageal disorders. 

Going deeper, the analysis at the family level demonstrated that *Prevotellaceae* (belonging to the Bacteroidetes order) was significantly higher during cancer progression. Within *Prevotellaceae*, the main genus noted was *Prevotella*. The dysbiosis state of the esophagus is generally expressed as a decrease in the Firmicutes/Bacteroidetes ratio and, in particular, in a decreased value of *Streptococcus*/*Prevotella* [32,33]. At the genus level, nevertheless, a similar abundance of *Prevotella* was observed during the progression of the disease, and instead there was a significantly increasing trend in *Alloprevotella*’s relative abundance. To date, no other studies have investigated the possible role of *Alloprevotella* in EAC development. 

To determine the specific bacterial composition related to EAC development, further analyses were performed comparing BE patients with patients who progressed, namely, those affected by dysplasia or cancer, and in our case LGD, HGD, and EAC together. The analysis of beta diversity comparing these two populations showed a markedly significant difference in microbial composition. Moving into the taxonomic level of bacteria genera, a higher abundance of *Bergeyella*, *Alloprevotella* (Bacteroidetes), and *Aggregatibacter* was observed. The increased amounts of *Alloprovetella* and *Aggregatibacter*’s relative abundance comparing BE and progressed patients reflect previous findings comparing each stage, suggesting the possible implication of these bacteria in cancer development. Conversely, some bacteria belonging to Proteobacteria decreased in progressed patients, in particular *Burkholderiaceae* and *Rhizobiaceae* families. As already mentioned, the so-called Type I microbiome, considered the normal composition of bacterial esophageal mucosa flora, was dominated by Gram-positive bacteria which gradually underwent a shift during carcinogenesis, leading to a higher Gram-negative abundance [15]. Our results suggest that on one hand, some Gram-negative bacteria increased during EAC development (*Alloprovetella* and *Aggregatibacter*), but on the other hand some Gram-negative bacteria decreased throughout the disease progression (in particular *Burkholderiaceae* and *Rhizobiaceae* families).

Collectively, our data, in accordance with the others’, suggest that the differences occurring in the esophageal microbiota during carcinogenesis were principally due to low-abundance bacteria rather than the most abundant bacteria. These low-abundance bacteria could act as a hallmark of cancer progression, like a signature of altered microbiota, and they may become a detection biomarker, similar to what has already been demonstrated in the improvement of colorectal cancer surveillance by introducing the detection of *F. nucleatum* [34]. Furthermore, it would be possible to combine different bacteria candidates to increase the performance of the diagnostic procedure [35].

Further analyses were performed in our cohort to predict the dependence of microbial features and the disease progression stages that could be used as a hallmark and to distinguish dysplastic and cancer patients from pre-cancerous stages. First, the associations between microbial factors and cancer development were determined with univariate binary logistic regression analysis. Successively, variables with statistically significant associations on univariate analysis were further included in a multivariate binary logistic regression model. Among the patients included in our cohort, six features identified as risk conditions were significantly related to cancer progression, as shown in the multivariate analysis. Although in esophageal samples no significant associations with EAC development were found among relative bacterial abundance, we set up further analyses considering the contribution of each class, order, family, or genus within a higher taxonomic level and, as a consequence, within the whole resident microbiome. The Firmicutes/Bacteroidetes ratio was already considered to be a good estimator of a healthy intestinal microbiome, for the value of the contribution of one compared to the other [24,26]. Likewise, the estimation of the genus ratio *Prevotella*/Bacteroides (both belonging to p_Bacteroidetes, c_Bacteroidia, and o_Bacteroidales) could potentially identify overweight or obese subjects who respond better to a low-calorie diet [25,26]. After both the univariate and multivariate analyses, six parameters defined as risk conditions were found to be significantly associated with cancer progression. The model, called REM-DT, was constructed using the set of microbial risk conditions and distinguished non-dysplastic BE from progressed patients (AUC = 0.92), with a negative predicted value of 96.9% and an accuracy of 85.0%. Interestingly, 96.2% of BE patients with a value of REM-DT lower than three were disease-free and only 3.8% progressed to low-grade dysplasia in a mean follow-up period of 48 months. The clinical impact of the proposed multiparametric score needs to be further investigated in a larger cohort of BE patients with a longer follow-up, but our data suggest it could sensibly improve patients’ management and ameliorate their condition. Studies have demonstrated the importance of improving easily controllable parameters such as weight, diet, and lifestyle habits to reduce EAC risk [36,37]. From this perspective, the administration of probiotics could be a potential strategy. Probiotics have already been suggested to ameliorate inflammation and GERD symptoms, as well as the altered obesity-related microbiome [38,39]. Liu et al. [40] performed a randomized prospective trial to study the effects of probiotics (a combination of *Bifidobacterium*, *Lactobacillus*, and *Enterococcus*) on gastrointestinal complications and nutritional status in postoperative patients with esophageal cancer. Bacteria such as *Lactobacillus* and *Bifidobacterium* are responsible for the immune response affecting pathogens, producing short-chain fatty acids such as lactic acid. Moreover, it has been shown that these bacteria could interact with stomach mucosal receptors, accelerating gastric emptying and relaxing the lower esophageal sphincter relaxation, one of the pathophysiological mechanisms of GERD [38,41]. The modifications of these parameters could, in turn, improve the gut microbiome in patients with esophageal cancer [42], but we still need robust data showing the ability of probiotics to permanently modify the resident esophageal microbiome.

Finally, the correlation between three significant microbial parameters used to define the REM-DT and the adherence to WCRF/AICR further confirms the role of diet and lifestyle habits in EAC prevention. 

This study had some limitations, including the low sample size, the necessity of validation in an independent and larger cohort with a more extended follow-up period, its retrospective nature, and the 16rRNA sequencing, which showed low power at the species level. Conversely, our study found its main power and strength in multiple aspects, including the focus on the European population, which could show more similar data to the real esophageal microbiota composition of BE patients commonly seen in Europe. Moreover, the development of a simple predictive score for HGD or cancer detection could fill the gap between speculative and quite complicated research in this field, and in clinical practice.

## 5. Conclusions

Collectively, our data, in accordance with others’, suggest that the differences occurring in esophageal microbiota during carcinogenesis are principally due to low-abundance bacteria rather than the most abundant bacteria. These low-abundance bacteria could act as a signature of altered microbiota. Here, we proposed a multiparametric test able to discriminate between pre-cancerous patients without dysplasia or cancer and those who progressed through dysplasia/cancer, and were potentially able to identify patients with a higher risk of developing cancer. The development of microbiome-based risk prediction models for some types of cancer, such as esophageal cancer, has opened new research avenues, demonstrating that the microbiome may be a valid non-invasive risk biomarker.

## Figures and Tables

**Figure 1 nutrients-15-02885-f001:**
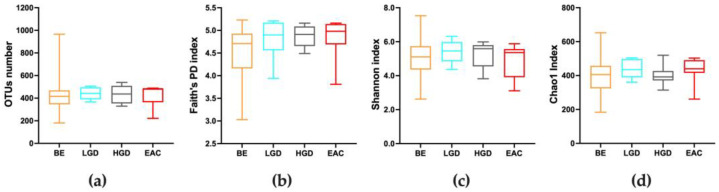
Quality of sequencing and alpha diversity measurements. (**a**) Number of OTUs in esophageal biopsies in all subjects according to each group considered. (**b**–**d**) Comparison of alpha diversity measures in esophageal microbiota among the groups considered for (**b**) Faith’s Phylogenetic Distance, (**c**) Shannon index, and (**d**) Chao 1 index. Box plots represent the median, interquartile range, and lower and minimum values. BE: patients with non-dysplastic BE; LGD: patients with low-grade dysplasia; HGD: patients with high-grade dysplasia; EAC: patients with esophageal adenocarcinoma.

**Figure 2 nutrients-15-02885-f002:**
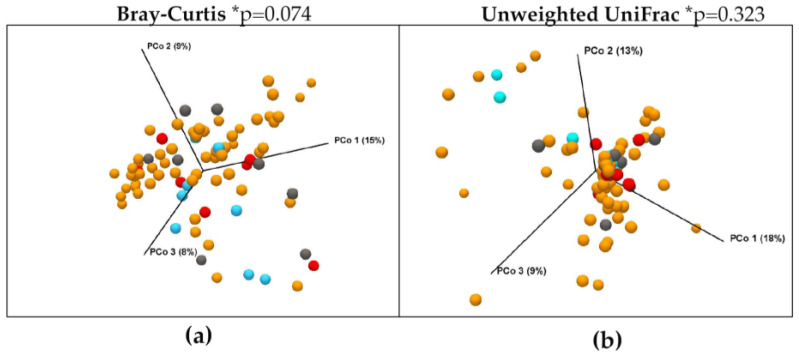
Principal coordinate analysis (PCoA) 3D plots of (**a**) Bray–Curtis and (**b**) Unweighted UniFrac in which samples were colored according to clinical outcome. Orange dots represent non-dysplastic BE patients; light-blue dots represent LGD patients; grey dots represent HGD patients; and red dots represent EAC patients. * PERMANOVA analysis of beta diversity.

**Figure 3 nutrients-15-02885-f003:**
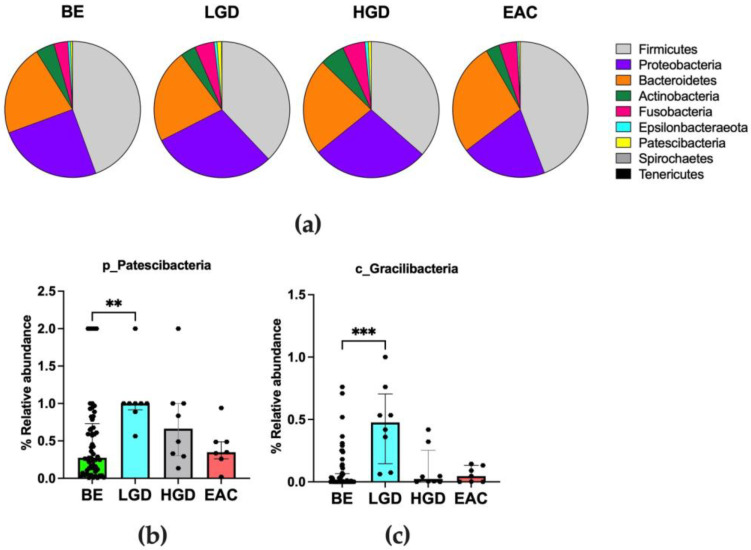
(**a**) Pie charts representing the relative abundance of the main phyla colonizing the distal esophageal tissues according to each group of patients considered. Data are shown as the median value. Phyla with a relative abundance higher than 0.5% are plotted. (**b**,**c**) Bacteria that significantly changed in esophageal microbiota during EAC. Post hoc analyses were annotated as ** *p* < 0.01, *** *p* < 0.001. BE: non-dysplastic BE; LGD: patients with low-grade dysplasia; HGD: patients with high-grade dysplasia; EAC: patients affected by esophageal adenocarcinoma.

**Figure 4 nutrients-15-02885-f004:**
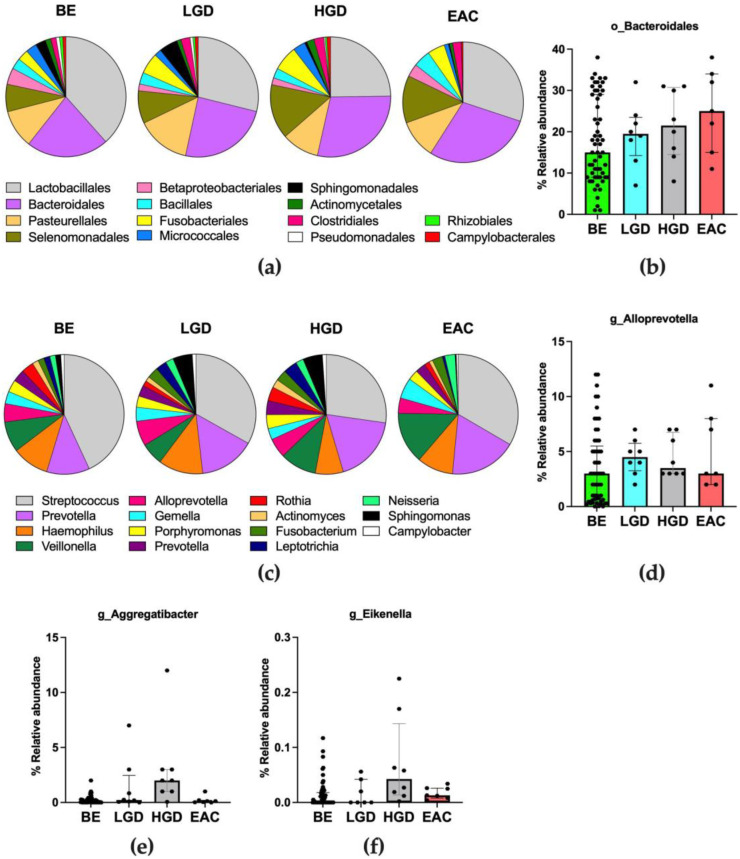
(**a**) Pie charts representing the relative abundance of the main orders colonizing the distal esophageal tissues according to each group of patients considered. Data are shown as the median value. Orders with a relative abundance higher than 0.5% are plotted. (**b**) Bacterial orders that significantly changed in esophageal microbiota during EAC. (**c**) Pie charts representing the relative abundance of the main genera colonizing the distal esophageal tissues according to each group of patients considered. Data are shown as the median value. Genera with a relative abundance higher than 0.5% are plotted. (**d**–**f**) Bacterial genera that significantly changed in esophageal microbiota during EAC. BE: non-dysplastic BE; LGD: patients with low-grade dysplasia; HGD: patients with high-grade dysplasia; EAC: patients affected by esophageal adenocarcinoma.

**Figure 5 nutrients-15-02885-f005:**
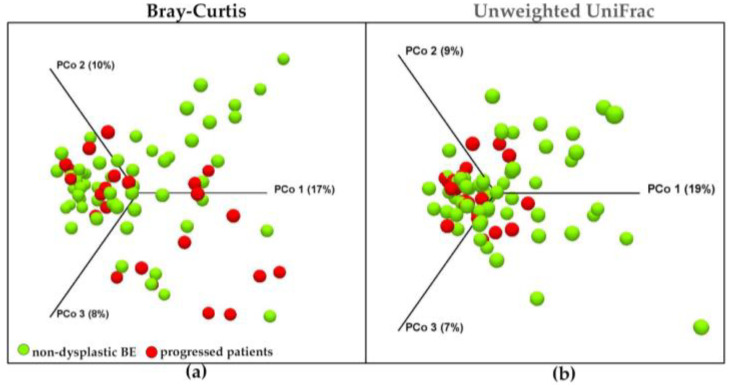
Principal coordinate analysis (PCoA) 3D plots of (**a**) the Bray–Curtis and (**b**) the Unweighted UniFrac. Each point represents a sample. Green dots represent non-dysplastic BE patients and red dots represent patients who progressed (LGD, HGD, and EAC together).

**Figure 6 nutrients-15-02885-f006:**
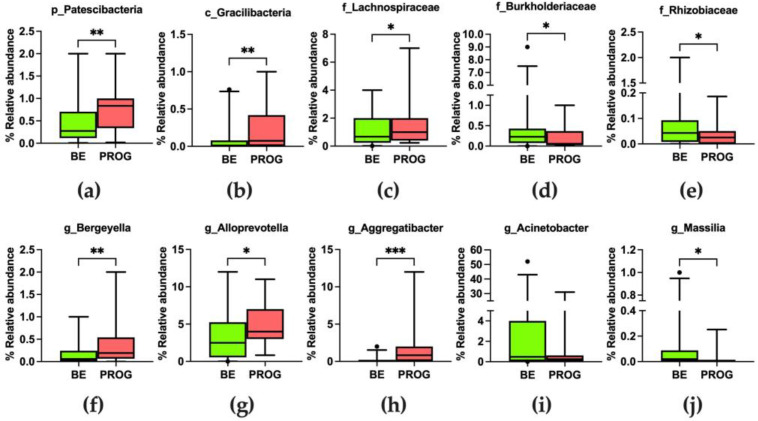
Comparison of bacterial abundance between non-dysplastic BE and progressed patients (defined as “PROG”) at different taxonomic levels. Data are represented as median, minimum to maximum values. Statistically significant differences in relative abundance at phylum (**a**), class (**b**), family (**c**–**e**), and genus (**f**–**j**) levels were analyzed with the Mann–Whitney U test and annotated as * *p* < 0.05, ** *p* < 0.01, *** *p* < 0.001.

**Figure 7 nutrients-15-02885-f007:**
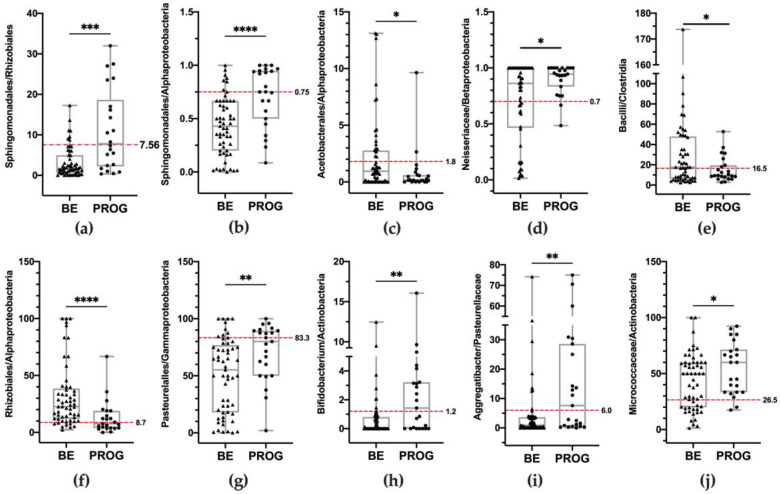
Comparison of the risk conditions in non-dysplastic BE and progressed patients (defined as “PROG”). The risk conditions were calculated as the ratio of relative abundance of (**a**) o_Sphingomonadales/o_Rhizobiales, (**b**) o_Sphingomoandales/c_Alphaproteobacteria, (**c**) o_Acetobacterales/c_Alphaproteobacteria, (**d**) f__Neiseriaceae/o_Betaproteobacteria, (**e**) c_Bacilli/c_Clostridia, (**f**) o_Rhizobiales/c_Alphaproteobacteria, (**g**) o_Pasteurellales/c_Gammaproteobacteria, (**h**) g_Bifidobacterium/c_Actinobacteria, (**i**) g_Aggregatibacter/f_Pasteurellaceae, and (**j**) f_Micrococcaceae/c_Actinobacteria. Boxplots show the median and interquartile range. Each patient belonging to the non-dysplastic BE group is represented by black triangles, and each patient belonging to the progression group is represented by black circles. The horizontal dotted red lines represent the cut-offs considered (the first or the third quartile of the whole data distribution, as appropriate) and indicated the level above or below which the neoplastic lesions were more frequent. Statistical analyses were performed using the Mann–Whitney U test and annotated as * *p* < 0.05, ** *p* < 0.01, *** *p* < 0.001, **** *p* < 0.0001.

**Figure 8 nutrients-15-02885-f008:**
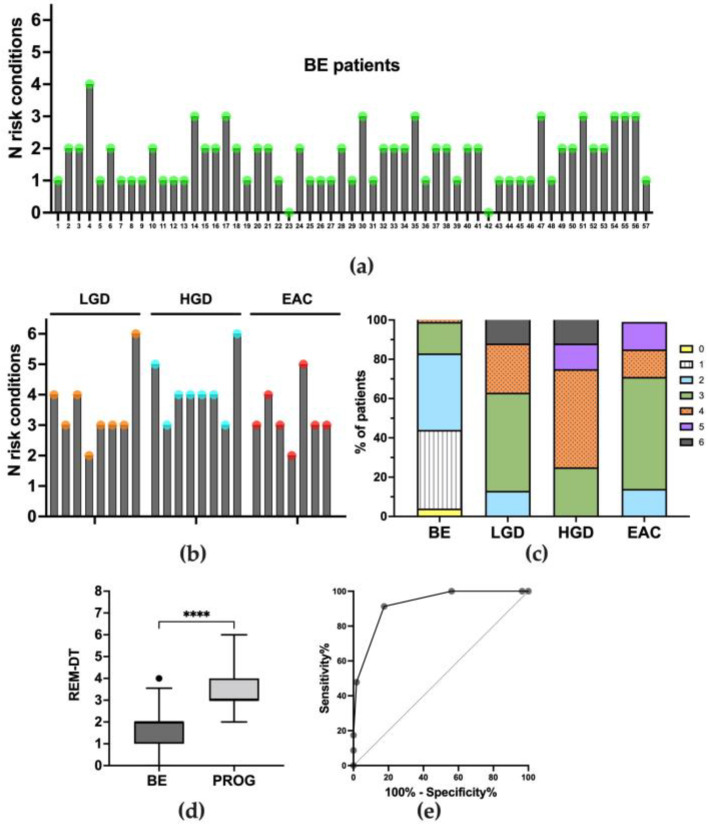
The number of risk conditions for each patient in (**a**) the BE group and (**b**) the progression group. (**c**) Prevalence of the number of risk conditions in each group studied. BE: patients with non-dysplastic Barrett’s esophagus; LGD: patients with low-grade dysplasia; HGD: patients with high-grade dysplasia; EAC: patients with esophageal adenocarcinoma. (**d**) Comparison of REM-DT score in BE patients and progression group. Statistical analysis was performed using the Mann–Whitney U test and annotated as **** *p* < 0.0001. (**e**) The ROC curve analysis shows the performance of our test to identify the progressor patients in our cohort.

**Figure 9 nutrients-15-02885-f009:**
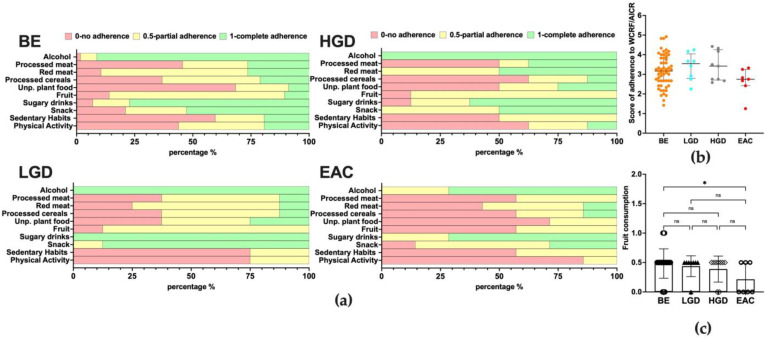
(**a**) Questionnaire results in study groups. Data are expressed as the percentages of patients. (**b**) The score of adherence for study groups. Each dot represents a patient. Data are expressed as the median and interquartile range. (**c**) Comparison of fruit consumption among study groups. Data are expressed as the median and interquartile range. Post hoc analyses were annotated as * *p* < 0.05 or not significant (ns).

**Figure 10 nutrients-15-02885-f010:**
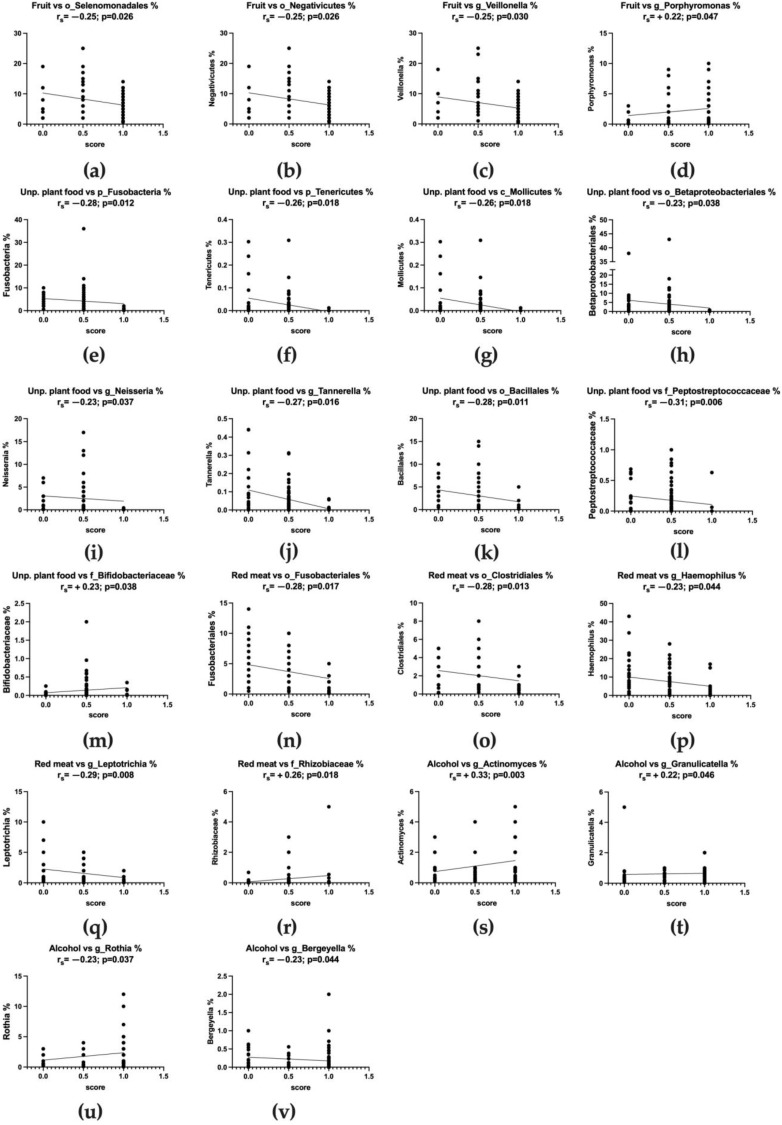
Correlation between questionnaire items and microbial parameters.

**Figure 11 nutrients-15-02885-f011:**
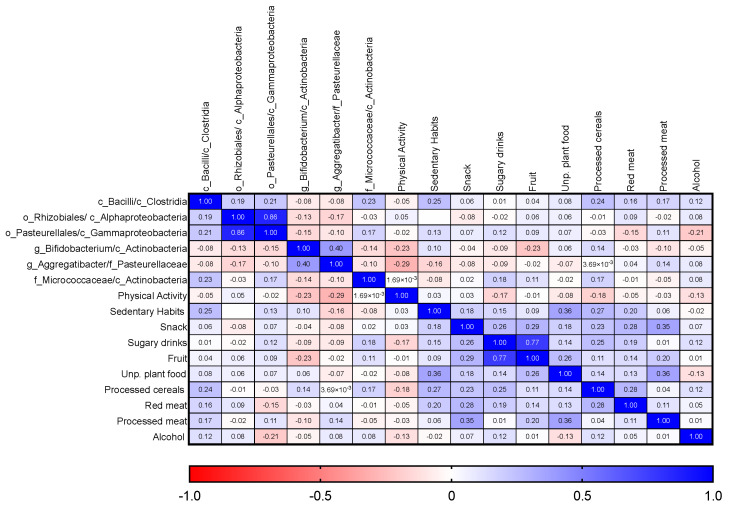
Correlation matrix between microbial parameters used in the calculation of REM-DT and questionnaire items.

**Table 1 nutrients-15-02885-t001:** WCRF/AICR recommendations utilized for the study.

WCRF/AICR Recommendation	Sub-Recommendations	Operationalization	Scoring
Body fatness: be as lean as possible without becoming underweight	(a) Ensure that body weight throughout childhood and adolescent growth projects toward the lower end of normal BMI range at age 21	Due to the already established role of both excessive BMI and waist circumference as independent risk factors for BE and EAC, this recommendation was not included in the questionnaire	NA
	(b) Maintain body weight within normal range from the age of 21		
	(c) Avoid weight gain and increase in waist circumference throughout adulthood		
Physical activity: be physically active as part of your everyday life	(a) Be moderately physically active, equivalent of brisk walking ≥30 min every day	Heavy manual job, or ≥2 h/week of vigorous physical activity, or ≥30 min of brisk walking every day	1
		15–30 min/day of brisk walking	0.5
		Less than 15 min/day of brisk walking	0
	(c) Limit sedentary habits such as watching television	≤1 h/day of sedentary activities	1
		Between 1 and 3 h/day of sedentary activities	0.5
		≥3 h/day of sedentary activities	0
Foods and drinks that promote weight gain: limit consumption of energy-dense foods and avoid sugary drinks	(a) Consume energy-dense foods sparingly	Less than one serving per week	1
		Between one and two servings per week	0.5
		More than two servings per week	0
	(b) Avoid sugary drinks	Less than 250 g of sugary drinks per week	1
		Between 250 g and 500 g of sugary drinks per week	0.5
		More than 500 g of sugary drinks per week	0
Plant foods: eat mostly foods of plant origin	(a) Eat ≥5 servings of non-starchy vegetables and/or fruits every day	≥5 servings per day	1
		Between 1 and 5 servings per day	0.5
		≤1 serving per day/not at all	0
	(b) Eat relatively unprocessed cereals and pulses with every meal	>1 serving per day	1
		1 serving per day	0.5
		<1 serving per day/not at all	0
	(c) Limit refined starchy foods	<1 serving per day/not at all	1
		1 serving per day	0.5
		>1 serving per day	0
Animal foods: limit intake of red meat and avoid processed meat	(a) People who eat red meat should consume <500 g/wk and very few, if any, processed meats	No red meat intake	1
		Red meat intake <500 g/wk	0.5
		Red meat intake ≥500 g/wk	0
		No processed meat intake	1
		<50 g/day processed meat intake	0.5
		≥50 g/day processed meat intake	0
Alcoholic drinks: limit alcoholic drinks	(a) If alcoholic drinks are consumed, limit consumption to ≤2 drinks/d for men and 1 drink/d for women	<2 drink/d (men) or <1 drink/d (women)	1
		2 drink/d (men) or 1 drink/d (women)	0.5
		≥2 drink/d (men) or ≥1 drink/d (women)	0
Preservation, processing, preparation: limit consumption of salt, avoid moldy cereals or pulses	(a) Avoid salt-preserved, or salty foods; preserve foods without using salt	Insufficient data available	NA
	(b) Limit consumption of processed foods with salt to ensure an intake <6 g/d of salt	Insufficient data available	NA
	(c) Do not eat moldy cereals or pulses	Not applicable to this population	NA
Dietary supplements: aim to meet nutritional needs through diet alone	(a) Dietary supplements are not recommended for cancer prevention	Insufficient data available	NA

**Table 2 nutrients-15-02885-t002:** Characteristics of patients according to each study group.

	BE	LGD	HGD	EAC	*p* Value
n	57	8	8	7	n/a
AGE, mean (±SD)	58.3 (±10.7)	57.9 (±7.8)	64.0 (±6.5)	62.3 (±12.6)	0.375 *
SEX: female, n (%)	8 (14)	0 (0)	1 (12)	0 (0)	n/a

SD: Standard deviation; n: number; * Kruskal–Wallis test; n/a: not applicable.

**Table 3 nutrients-15-02885-t003:** The univariate logistic and multivariate binary logistic regression analyses according to the risk conditions for EAC progression.

			Univariate			Multivariate	
Risk Conditions	BE n = 57	PROG n = 23	OR	95% CI	p	p	95% CI
Sphingomondales/Rhizobiales ≥7.56 (Q3)	8 (14.0)	12 (52.2)	6.68	2.21 to 20.24	0.001	n.s.	/
Sphingomonadales/ Alphaproteobacteria ≥0.75 (Q3)	7 (12.3)	12 (52.2)	7.79	2.50 to 24.32	<0.001	n.s.	/
Acetobacterales/Alphaproteobacteria ≤1.8 (Q3)	39 (68.4)	21 (91.3)	4.85	1.02 to 22.93	0.046	n.s.	/
Neisseriaceae/Betaproteobacteriales ≥0.70 (Q3)	38 (66.7)	22 (95.6)	11.00	1.38 to 87.90	0.024	n.s.	/
Bacilli/Clostridia ≤16.5 (Q2)	25 (43.9)	16 (69.6)	2.92	1.04 to 8.22	0.041	0.023	0.030 to 0.397
Rhizobiales/Alphaproteobacteria ≤8.7 (Q1)	8 (14.0)	12 (52.2)	6.68	2.21 to 20.24	0.001	0.003	0.106 to 0.491
Pasteurellales/Gammaproteobacteria ≥83.3 (Q3)	10 (17.5)	11 (47.8)	4.22	1.45 to 12.25	0.008	0.021	0.037 to 0.427
Bifidobacterium/Actinobacteria ≥1.2 (Q3)	8 (14.0)	12 (52.2)	6.68	2.21 to 20.24	0.001	<0.001	0.174 to 0.537
Aggregatibacter/Pasteurellaceae ≥6.0 (Q3)	8 (14.0)	12 (52.2)	6.68	2.21 to 20.24	0.001	0.037	0.012 to 0.387
Micrococcaceae/Actinobacteria ≥26.5 (Q1)	39 (68.4)	21 (91.3)	4.85	1.02 to 22.93	0.047	0.002	0.094 to 0.407

BE: non-dysplastic BE patients; PROG: progressed patients (LGD, HGD, EAC together). Odds ratios (OR) and 95% confidence intervals (CIs) measure the association between ratio values and disease progression. The cut-offs were considered to be the Q1 or Q3 of the whole data distribution and indicated the level above (>or equal) or below (<or equal), in which neoplastic lesions were significantly more frequent than BE. n.s.: *p* value is not statistically significant (*p* > 0.05).

## Data Availability

The data presented in this study are available on request from the corresponding author.

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
