# Peer review of "Resident Esophageal Microbiota Dysbiosis Correlates with Cancer Risk in Barrett’s Esophagus Patients and Is Linked to Low Adherence to WCRF/AICR Lifestyle Recommendations"

_nutrients, 2023, doi:10.3390/nu15132885_

Round 1

Reviewer 1 Report

This is an interesting paper as there is a hot research topic on the role of esophageal microbiota on Esophageal adenocarcinoma. From this manuscript, authors evaluated the association between esophageal microbiota and dietary/lifestyle habits during EAC development. This study has certain value in exploring Esophageal adenocarcinoma development. However, I think this manuscript needs to be partially revised.

1.     the author mentioned “This work aimed to investigate esophageal microbiota in patients with different esophageal lesions, and evaluated the association between its main features and dietary/lifestyle habits during EAC development. The main goal was to identify the most representative microbial parameters and to develop a multiparametric score able to predict the risk of progression toward EAC in BE patients to improve patient management.” However, I did not see any conclusions regarding these purposes in the abstract

2.     In Introduction, the author should mention the possible specific resident gastroesophageal microbiota.

3.     In the discussion, the author should also mention the advantages and limitations of this study

4.     What confuses me about this research is that I haven't seen the specific conclusions of the article.

Author Response

Response to Reviewer 1 Comments

Point 1: the author mentioned “This work aimed to investigate esophageal microbiota in patients with different esophageal lesions, and evaluated the association between its main features and dietary/lifestyle habits during EAC development. The main goal was to identify the most representative microbial parameters and to develop a multiparametric score able to predict the risk of progression toward EAC in BE patients to improve patient management.” However, I did not see any conclusions regarding these purposes in the abstract

Response 1: Thank You very much for your suggestion. We rephrased the abstract adding a more detailed conclusion while remaining within the 250 words as required by the journal guidelines. From line 39, “The presence of the six significant microbial features with multivariate analysis was used to develop a multiparametric score (Resident Esophageal Microbial Dysbiosis Test) to predict the risk of progression toward EAC. Finally, the diagnostic ability of the test and its discrimination threshold able to identify dysplastic/cancer patients was demonstrated. “

Point 2: In Introduction, the author should mention the possible specific resident gastroesophageal microbiota.

Response 2: Thank You for your comments, we appreciate it. An overview of the resident esophageal microbiota was added. In line 79, “Ever since the early studies investigating esophageal microbiome composition, it was clear that Streptococcus, Prevotella, and Veillonella were the most represented genera in order of abundance. The composition could be influenced by age, drugs (especially proton pump inhibitors), and dietary habits [20-22]. The healthy microbial composition, principally by Gram-positive bacteria (Firmicutes phylum) was defined as Type I in the work of Yang and colleagues [15], suggesting the shift into a Type II during esophageal disease. Type II is dominated by Gram-negative taxa in patients with both GERD and BE.”

Point 3: In the discussion, the author should also mention the advantages and limitations of this study

Response 3: Thank You for your suggestion. Line 655 reported the limitations of the study “the low sample size, the necessity of validation in an independent and larger cohort with a more extended follow-up period, its retrospective nature, and the 16rRNA sequencing that shows low power at the species level”. Moreover, the advantages of our study are discussed from line 658 “the focus on the European population which could show more similar data to the real esophageal microbiota composition of BE patients commonly seen in Europe. Moreover, the development of a simple predictive score for HGD or cancer detection could fill the gap between speculative and quite complicated research in this field and clinical practice”.

Point 4: What confuses me about this research is that I haven't seen the specific conclusions of the article.

Response 4: Thank You for your useful comment. We added a final section in line 644, including our conclusions was added: “Collectively, our data, in accordance with the others, suggests that the differences occurring in esophageal microbiota during carcinogenesis were principally due to low-abundance bacteria rather than the most abundant bacteria. These low-abundance bacteria could act as a signature of altered microbiota. Here we proposed a multiparametric test able to discriminate pre-cancerous patients without dysplasia or cancer from those who progressed through dysplasia/cancer and potentially able to identify patients with a higher risk to develop cancer. The development of microbiome-based risk prediction models for some types of cancer, such as esophageal cancer, has opened new research avenues, demonstrating that the microbiome may be a valid non-invasive risk biomarker.”

Reviewer 2 Report

Well planned, conducted study. The result are good and convincing and discussion is reasonable. The overall impression about the work is very good.

It will be nice if the authors can make a suggestion whether increased intake of of Lactobacillus or other suitable probiotic will help to reduce the incidence of EAC. 

Author Response

Response to Reviewer 2 Comments

Point 1: Well planned, conducted study. The result are good and convincing and discussion is reasonable. The overall impression about the work is very good.

It will be nice if the authors can make a suggestion whether increased intake of Lactobacillus or other suitable probiotic will help to reduce the incidence of EAC.

Response 1: Thank You very much for your comments. We really appreciate your positive feedback and suggestions.

The administration of Lactobacillus and/or other probiotics to reduce the incidence of EAC could be an interesting point of discussion. In the manuscript, from line 634, we underline what is already known regarding the administration of probiotics and their potential effect on the gut microbiome in patients affected by esophageal cancer.

“Studies demonstrate the importance of improving easily controllable parameters such as overweight, diet, and lifestyle habits to reduce EAC risk [38,39]. In this view, also the administration of probiotics could be a potential strategy. Probiotics were already suggested to ameliorate inflammation and GERD symptoms as well as the altered obesity-related microbiome [40,41]. Liu et al [42], are performing a randomized prospective trial to study the effects of probiotics (a combination of Bifidobacterium, Lactobacillus, and Enterococcus) on gastrointestinal complications and nutritional status in postoperative patients with esophageal cancer. Bacteria such as Lactobacillus and Bifidobacterium are responsible for the immune response toward pathogens, producing short-chain fatty acids, such as lactic acid. Moreover, it was shown that these bacteria could interact with stomach mucosal receptors accelerating gastric emptying and relaxing the lower esophageal sphincter relaxation, one of the pathophysiological mechanisms of GERD [40,43]. The modifications of these parameters could in turn improve the gut microbiome in patients with esophageal cancer [44], but we still need robust data showing the ability of probiotics to permanently,  modify the resident esophageal microbiome.”

Reviewer 3 Report

The authors have very well explained the manuscript. I have no questions.

Minor English changes required

Author Response

Response to Reviewer 3 Comments

Point 1:

The authors have very well explained the manuscript. I have no questions.

Comments on the Quality of English Language

Minor English changes required

Response 1: Thank You for reviewing our work. We greatly appreciate your positive comments regarding our manuscript. The required minor English editing was done.
